# Selection of Prebiotic Molecules in Amphiphilic Environments

**DOI:** 10.3390/life7010003

**Published:** 2017-01-07

**Authors:** Christian Mayer, Ulrich Schreiber, María J. Dávila

**Affiliations:** 1Institute of Physical Chemistry, CENIDE University of Duisburg-Essen, 45141 Essen, Germany; 2Department of Geology, University of Duisburg-Essen, 45141 Essen, Germany; ulrich.schreiber@uni-due.de (U.S.); maria.davila@uni-due.de (M.J.D.)

**Keywords:** origin of life, selection, accumulation, prebiotic chemistry, molecular evolution, vesicles

## Abstract

A basic problem in all postulated pathways of prebiotic chemistry is the low concentration which generally is expected for interesting reactants in fluid environments. Even though compounds, like nucleobases, sugars or peptides, principally may form spontaneously under environmental conditions, they will always be rapidly diluted in an aqueous environment. In addition, any such reaction leads to side products which often exceed the desired compound and generally hamper the first steps of a subsequent molecular evolution. Therefore, a mechanism of selection and accumulation of relevant prebiotic compounds seems to be crucial for molecular evolution. A very efficient environment for selection and accumulation can be found in the fluid continuum circulating in tectonic fault zones. Vesicles which form spontaneously at a depth of approximately 1 km present a selective trap for amphiphilic molecules, especially for peptides composed of hydrophilic and hydrophobic amino acids in a suitable sequence. The accumulation effect is shown in a numeric simulation on a simplified model. Further, possible mechanisms of a molecular evolution in vesicle membranes are discussed. Altogether, the proposed scenario can be seen as an ideal environment for constant, undisturbed molecular evolution in and on cell-like compartments.

## 1. Introduction

In past research dealing with the origin of life, numerous natural sources of prebiotic chemistry have been identified [1]. This includes potential reactions in the early terrestrial atmosphere [2,3], geothermal chemistry in the Earth’s crust [4,5] as well as extraterrestrial sources [6,7,8]. In practically all these cases, the interesting molecular species are formed in very low quantities and in an extremely diluted state. Often, they are also accompanied by large varieties and huge amounts of side products which generally are meaningless or even obstructive for further steps of molecular evolution.

So basically, two conditions had to be fulfilled in order to allow the primary steps of an evolution process: (i) the selection and (ii) the accumulation of prebiotic compounds which formed the original basis for life. Among those two problems, accumulation seems to be the easier one to solve. In the simplest case, a phase transition (e.g., the evaporation or the freezing of water) leads to the precipitation of dissolved components. The degree of accumulation during such a step depends on the original solubility and may easily increase the concentration of a given species over several orders of magnitude.

The condition of molecular selection is much harder to fulfill. Side products often exhibit similar molecular properties as the “desired” component. Therefore, it is not easy to imagine a physical process which separates the tiny amount of relevant product from the large amount of waste. In many cases, a complex mixture of undesired components may even precipitate and cause a dead-end scenario for the local evolution process. This event has been referred to as the *asphalt problem* in the literature [9] and presents a serious drawback for many proposed mechanisms of molecular evolution.

A possible solution to both problems is a natural environment which acts in some respect similar to a chromatographic column: the system of channels and cavities being formed by tectonic fault zones in the continental crust (Figure 1).

The fluid content, primarily consisting of a mixture of liquid water and gaseous (g) or supercritical (sc) carbon dioxide acts like a mobile phase, while organic precipitates and mesoscopic structures (formed at depths where the sc-g transition occurs) represent the relatively immobile counterpart. Regarding the selection and accumulation of prebiotic compounds, this system combines some intriguing properties:
(a)It is constantly supplied with a huge amount and a huge variety of hydrothermally formed compounds from geological sources.(b)It is well protected against potentially destructive influences such as variable weather conditions or UV radiation.(c)It provides constant conditions over long periods of time (such as thousands of years), hereby allowing for undisturbed molecular evolution.(d)It generates structures which are highly selective for amphiphilic compounds such as lipids or peptides composed of hydrophilic and hydrophobic amino acids in a suitable sequence.(e)The system has access to the surface allowing the products of the molecular and structural evolution to spread out into new environments, especially on and near the terrestrial surface.


With these advantages in mind, we want to propose a possible scenario of selection, accumulation and evolution which starts with hydrothermal chemistry in a highly diluted state, includes the formation of mesoscopic structures which absorb and accumulate a certain fraction of intermediate products, and allows for highly productive periodic processes.

## 2. Tectonic Fault Zones—A Description of the Environment

Tectonic fault zones consist of a huge network of interconnected cracks and cavities [10,11]. Well known examples for tectonic fault zones are low-temperature hydrothermal systems with extensive gas flux in the continental crust. Their fluid content includes supercritical and subcritical water as well as supercritical and subcritical gases, primarily carbon dioxide. In addition, they are constantly supplied with organic building blocks by hydrothermal sources. An example for such a process is the synthesis of amino acids which is very likely to occur under the given circumstances [12,13,14]. Regarding the reaction conditions, tectonic fault zones offer a large range of temperatures and pressures depending on the depth of the reaction environment. Below a depth of approximately −1 km (with T > 304 K and *p* > 74 bar), the carbon dioxide will necessarily exist in the supercritical state [15]. At levels above −8 km, water is expected to occur in the liquid state. In between, both components together necessarily form a two-phase system offering a non-polar phase (supercritical carbon dioxide saturated with water) and a polar one (liquid water saturated with carbon dioxide). This two-phase system provides excellent conditions for synthetic organic chemistry, especially for the formation of amphiphilic compounds at the phase boundary. At the upper limit of this area (near −1 km), the rising carbon dioxide undergoes the phase transition from the supercritical to the gaseous state. At this point, the ability of the carbon dioxide to act as a solvent will drastically decrease and most of the less polar or amphiphilic organic constituents will precipitate. This region can be seen as an accumulation zone for such compounds, leading to locations with elevated concentrations of interesting prebiotic building blocks.

## 3. Vesicle Formation

Recently, we established a mechanism of vesicle formation which is expected to occur in tectonic fault zones in the presence of water and CO_2_ [11]. At a depth of approximately −1 km, pressure and temperature conditions induce a local phase transition between supercritical CO_2_ (scCO_2_) and subcritical gaseous CO_2_ (gCO_2_). The uncertainty regarding the precise depth derives from the variable composition of the gas mixture and from the fact that the density of the liquid filling of the vertical channels depends on the volume fraction of CO_2_ bubbles which can vary over time. Various amphiphilic products, either formed by Fischer-Tropsch like chemistry under oxidation of the terminal end group of aliphatic chains or by condensation reactions with glycerol [16], are expected to accumulate at this point due to the solubility drop in CO_2_ and the presence of large transition-induced interfaces.

With additional periodic pressure variations resulting from tidal influences or geyser phenomena [17], a cyclic process can occur in which the transition scCO_2_→gCO_2_ induces the formation of water droplets covered by a monolayer of amphiphilic compounds [18]. When migrating through the interface to the aqueous domain (which by itself is covered by a layer of amphiphiles), the droplets turn into vesicles with a bilayer membrane [19]. Being thermodynamically unstable, the vesicles are expected to disintegrate and release their organic contents into the bulk water phase over time. During the transition gCO_2_→scCO_2_, the organic constituents and the water again become soluble in the CO_2_ phase and the cycle can start again [11].

This leads to a continuous process of repetitive compartmentalization and subsequent remixing of water-soluble organic components which occurs at a well-defined depth inside the Earth’s crust. The compartments would likely have lifetimes between minutes and days and allow for extended reactions to occur under variable internal conditions. The transient state of encapsulation is especially interesting as there are experimental indications that vesicles facilitate the formation of peptides [20,21] and cause their selective immobilization [22]. The described process of vesicle formation may also solve the concentration problem (which basically consists in the constant dilution of water soluble organic compounds): while an increased concentration of prebiotic molecules is already expected at the accumulation zone, the process of droplet formation leads to further accumulation of prebiotic molecules due to the low solubility of organic compounds in gCO_2_ [11,23].

Experimental evidence for the vesicle formation was gained from systematic studies using a high-pressure cell filled with water/scCO_2_ in the presence of phospholipid as a model amphiphile. A pressure cycling results in the formation of vesicles which were characterized by optical microscopy and field-gradient nuclear magnetic resonance spectroscopy (PFG-NMR) [11]. The size of the vesicles differs between one and 60 µm. Some of the vesicles obviously show multilamellar structure and contain internal vesicles, both features possibly deriving from the fusion of droplets in the gas phase (Figure 2). The vesicle structure formed by bilayers of amphiphilic molecules could further be supported by PFG-NMR measurements which showed the presence of encapsulated water molecules inside membranes of relatively low permeability [11].

## 4. Accumulation and Selection of Molecules in Vesicle Membranes

With the described mechanism of vesicle formation inside the tectonic fault zone, we can postulate the occurrence of a layer with a relatively high concentration of vesicles at a depth of 1 km at temperatures between 310 and 350 K and pressures near 70 bar. In the following, this region will be referred to as the “vesicle zone”.

In some respect, the vesicles in this layer can be compared to a static phase of a chromatographic column: the bilayer membranes of the dispersed vesicles specifically absorb molecules with amphiphilic properties, thereby selecting and accumulating them. Molecules which are capable of integrating into the bilayer structure will combine with the present collection of amphiphiles, forming a heterogeneous membrane of increasing complexity. Even though individual vesicles will disintegrate after a given period of time, bilayer-compatible molecules will remain in the vesicle zone as they have the potential to escape into the bulk phase and to rapidly integrate into another intact vesicle. Consequently, the vesicle zone forms a static environment, even though individual vesicles may be mobile or unstable.

Within this environment, potential prebiotic molecules will undergo chemical reactions and will change their properties. As soon as they lose their amphiphilic character, they will also lose their ability to remain in the vesicle zone. If they turn into a hydrophilic specimen, they are likely to be eluted by the aqueous phase. If instead they gain hydrophobic properties, their solubility in the supercritical carbon dioxide phase will increase and they will be eluted in those periods of time with increased pressure when scCO_2_ occurs.

The potential of this selection mechanism is especially high in the case of peptides which are being formed from racemic amino acids under hydrothermal conditions [24]. The selectivity for hydrophobic peptides has already gained early recognition [22]. However, the selection effect may even be stronger for those peptides which, with a suitable composition and sequence of amino acids, are amphiphilic and reflect the spatial distribution of hydrophilicity and hydrophobicity of the bilayer membrane.

Preliminary experiments have shown that peptides are being formed from amino acids in the same environment where the vesicle cycle occurs. Under these circumstances, an accumulation of amphiphilic peptides will occur according to the scheme shown in Figure 3.

Basically, three different types of oligomers can be classified according to their behavior in the system:
(1)Hydrophilic amino acids and primarily hydrophilic oligopeptides (left in Figure 3) will primarily remain dissolved in the aqueous phase. Here, they are part of a series of connected condensation–hydrolysis equilibria where oligomers with increasing chain length occur in decreasing concentration. Regarding the long-term contribution of this fraction of molecules, it can be considered as a material reservoir for amphiphilic amino acids. Regarding the formation of longer chains, this fraction suffers from the *water problem* [9].(2)Hydrophobic amino acids and primarily hydrophobic oligopeptides (right in Figure 3) will have a higher tendency to dissolve in the scCO_2_ phase. Even though its solubility in scCO_2_ may be very small in terms of numbers, the huge amount of carbon dioxide passing by at the water interface can carry a substantial amount over time. With the high mobility of the supercritical medium, these fractions are likely to be separated and to end up at positions where carbon dioxide is undergoing transition towards the gaseous state. The removal of this fraction is important as it prevents the formation of an insoluble hydrophobic residue which otherwise could block further reactions. Without a given selectivity, this product mixture is likely to gain in complexity and to lead to a dead end which has been referred to as the *asphalt problem* by Steven Benner [9].(3)Peptides formed from a combination of hydrophilic and hydrophobic amino acids may develop a tendency to integrate into the bilayer membranes of existing vesicles (center in Figure 3). This requires a stretch of hydrophobic amino acids which is long enough to fit into the aliphatic chain region of the bilayer together with at least one hydrophilic part which reaches into the polar head group region. It would be incorporated in the bilayer membrane by the hydrophobic effect: any dislocation of the peptide towards the inside or the outside of the vesicle would result in a steep increase of the free energy of the system. In this state, the peptide is partially protected against hydrolysis as the access of water molecules to the peptide bonds is hampered.


As a consequence of these three mechanisms, amphiphilic peptides will be selected and accumulate in the vesicle forming region. It has to be noted at this point that all amino acids are expected to occur in racemic mixtures. Therefore, the conformational structure of the resulting oligopeptides strongly differs from the biogenic versions, for example in the lack of their tendency to form alpha helices. Nevertheless, their basic affinity towards the bilayer region based on the amino acid sequence will be similar to existing biogenic versions. In fact, such a selected prebiotic oligopeptide could resemble the typical peptide anchor region of a recent membrane bound protein. Based on this hypothesis, one could assume that those anchor regions may be the oldest portions of functional proteins.

## 5. Numeric Simulation of Peptide Formation and the Accumulation Process

The rate of peptide formation can be visualized in a numeric calculation using a number of simple preconditions. We assume that the rate constants for the condensation reaction k_c_ as well as the one for hydrolysis of a single peptide bond k_h_ are equal for all amino acids and all peptides. Limiting the maximum length of peptides to six amino acid units, the following set of coupled differential equations is obtained:
(1)$$\frac{dc_{1}}{dt}=2\;k_{h}\;\left(c_{2}+c_{3}+c_{4}+c_{5}+c_{6}\right)-2\;k_{c}c_{1}\left(2c_{1}+c_{2}+c_{3}+c_{4}+c_{5}\right)$$
(2)$$\frac{dc_{2}}{dt}=2\;k_{c}c_{1}^{2}+2\;k_{h}\left(c_{3}+c_{4}+c_{5}+c_{6}\right)-2\;k_{c}c_{2}\left(c_{1}+2c_{2}+c_{3}+c_{4}\right)-k_{h}c_{2}$$
(3)$$\frac{dc_{3}}{dt}=2\;k_{c}c_{1}c_{2}+2\;k_{h}\left(c_{4}+c_{5}+c_{6}\right)-2\;k_{c}c_{3}\left(c_{1}+c_{2}+2c_{3}\right)-2k_{h}c_{3}$$
(4)$$\frac{dc_{4}}{dt}=2\;k_{c}c_{1}c_{3}+2\;k_{c}c_{2}^{2}+2\;k_{h}\left(c_{5}+c_{6}\right)-2\;k_{c}c_{4}\left(c_{1}+c_{2}\right)-3k_{h}c_{4}$$
(5)$$\frac{dc_{5}}{dt}=2\;k_{c}c_{1}c_{4}+2\;k_{c}c_{2}c_{3}+2\;k_{h}c_{6}-2\;k_{c}c_{5}c_{1}-\;4k_{h}c_{5}$$
(6)$$\frac{dc_{6}}{dt}=2\;k_{c}c_{1}c_{5}+2\;k_{c}c_{2}c_{4}+2\;k_{c}c_{3}^{2}-5k_{h}c_{6}$$
where c_n_ stands for the concentration of a peptide formed from n amino acid units (c_1_ being the concentration of single amino acids). The various pre-factors either account for the fact that a reaction can occur at two or more positions of the molecule or that two molecules are participating in the reaction.

The equilibrium constant of a single peptide bond formation is then given by the relation between k_c_ and k_h_. Preliminary experiments of amino acid mixtures treated in a high-pressure cell (*p* = 70 bar, T = 350 K) and analyzed by NMR show that a value of k_c_/k_h_ = 0.1 is plausible. This is expressed by the fact that the average concentrations of the amino acids c_1_ and the average concentration of dipeptides c_2_ in thermal equilibrium exhibit an approximate relation of 10:1 which, in a kinetic interpretation of the equilibrium constant, should also roughly correspond to the relation k_c_/k_h_. Further, a starting concentration of c_1_ = 1 a.u. is assumed for the amino acids (c_n_(0) for *n* > 1 is set to zero). Under these conditions, the differential Equations (1–6) can be developed over time to yield time dependent concentrations of the amino acids (c_1_) as well as of oligopeptides (c_n_) up to hexapeptides (c_6_). The result is shown on a logarithmic scale in Figure 4.

While the amino acid concentration is only slightly reduced, all peptide concentrations grow rapidly at the beginning and level off at their corresponding equilibrium values. In the first approximation, the final levels of c_1_, c_2_, c_3_, c_4_, c_5_, and c_6_ settle near the values 10^0^, 10^−1^, 10^−2^, 10^−3^, 10^−4^, and 10^−5^ a.u. for the limit of t→∞ (Figure 4b). The increasingly low level of the concentrations for peptides with higher chain lengths is basically the numeric representation of the *water problem* [9].

In the second step, we introduce the presence of vesicles in the reaction mixture. We assume that a certain fraction of the hexapeptides is amphiphilic, i.e., it consists of primarily hydrophilic amino acids at one end and primarily hydrophobic acids on the other. With the given selection of hydrothermally formed amino acids, we estimate that approximately 30% of the hexapeptides will fulfill this condition. We assume that this fraction of hexapeptides will integrate into the vesicle membrane with an equilibrium constant of K_c_ = k_in_/k_out_ = 1000, a value which is plausible for an amphiphile of this kind. Inside the membrane, it is likely to be protected against hydrolysis. The resulting plot for the concentration of the integrated hexapeptide fraction c_6_(i) is shown in Figure 5 (red line).

Initially, the given amphiphilic fraction of 30% of all hexapeptides is rapidly integrated, leading to a 7:3 ratio between c_6_ and c_6_(i) (c_6_ stands for the residual hexapeptide in solution). Over time, the non-amphiphilic fraction of hexapeptides will undergo hydrolysis and will deliver raw material for new (and then potentially amphiphilic) hexapeptides. For that reason, the fraction of integrated hexapeptide keeps growing and soon exceeds the values of c_6_ and c_5_ (Figure 5b). Depending on the integration constant k_in_/k_out_ and on the overall amount of vesicles, it can easily reach significant levels. This result may be a rough approximation of the real conditions, but it shows the actual power of the given selection mechanism combined with hydrothermal polymer formation.

## 6. Possible Mechanisms of Vesicle-Induced Molecular Evolution

Looking at the mechanism of peptide selection in detail, one can differentiate between three possible criteria favoring different types of amphiphilic peptides:
(1)**Integration**. In analogy to biological counterparts, this effect could also be named a “parasitic” behavior as the advantage is limited to the peptide itself. Amphiphilic peptides with an amphiphilicity profile fitting into the bilayer cross section integrate into the membrane. Thereby, they gain an individual selective advantage as they are partially protected against hydrolysis and as they become less easily eluted from the vesicle zone. For this selection criterion, the effect on the stability of the vesicle is negligible.(2)**Stabilization**. In this case, amphiphilic peptides are being protected by vesicles but in turn also stabilize the vesicle structure. Again, in analogy, one could categorize this mutual advantage as “symbiotic”. Very efficient candidates for species causing vesicle stabilization could be peptides consisting of a long hydrophobic part with two hydrophilic ends. In this case, the peptide could fully integrate into the bilayer and connect the two hydrophilic layers of the membrane in analogy to a bola-amphiphile [25]. Such bola-amphiphiles lead to a significantly higher thermal stability of the surrounding bilayer membrane and occur in thermophilic organisms. With increased stability, the vesicles gain an advantage from hosting the peptide molecules while the latter in turn enjoy an increased degree of protection. Altogether, this symbiotic effect would act more efficiently than the parasitic integration effect alone.(3)**Function**. This part of the possible evolution process is the most speculative, but also the most exciting one. The potential is obvious: peptides with new functions beyond a simple mechanical stabilization may develop. A possible example may be an ion channel based on a trans-membrane polypeptide [26]. In the vesicle cycle, the vesicles are generated with a natural ionic concentration gradient across the membrane which destabilizes their structure by the resulting osmotic stress. An ion channel could cause a rapid relaxation of this gradient, leading to a longer vesicle lifetime and resulting in a corresponding selection advantage for the channel-forming peptide. Other selected functions could include the ability to bind to a specific surface, to adopt a specific charge, to selectively bind to other vesicles, or even to make the membrane permeable for specific molecules. More complex versions of the peptides could even make use of the concentration gradient as a source of free energy. In any case, such functional peptides could be seen as initial primitive predecessors of functional membrane proteins.


Of course, the described mechanism of peptide selection misses an important precondition for actual molecular evolution: the capability for identical reproduction. However, the mechanism of selection by itself can be quite efficient as long as the pool of molecules to choose from is large enough and exists over long periods of time. Both conditions are definitely fulfilled for peptides in hydrothermal sources. Even if the “evolution” process is driven by selection only (and if a possible template effect is excluded), it could become remarkably complex: on the integration of selected peptides, the vesicles may change their affinity towards other amphiphilic molecules, resulting in a change of the membrane composition. This in turn may affect the selection criterion for further peptides, and so on (Figure 6).

With the integration of amphiphilic peptides, the vesicle membrane will form a very specific template which again influences the selection of further peptides. Therefore, one could regard the amphiphilicity profile of the membrane as something like a primitive memory for molecular structure (i.e., the sequence of hydrophilic and hydrophobic amino acids) which evolves and optimizes over time. Furthermore, as a principal rule, all molecules which are of an advantage for the formation of stable vesicles will remain in the vesicle zone and therefore accumulate in this environment. 

The result of this “evolution” process would be optimized, stable and maybe functional vesicles. Even though the majority of the vesicles is immobilized in a cycle of formation and destruction at the vesicle zone, a small fraction of especially stable vesicles could be transported to the terrestrial surface and into different local environments (e.g., sediment basins, ponds). Being thermally stable, they could survive over longer periods of time and could form versatile compartments for further developments in molecular evolution. Experimentally, the increased stability of these vesicles will be tested in a two-phase environment formed by water and supercritical carbon dioxide. Artificial pressure cycling induces the periodic formation of vesicles in the aqueous phase. Without further stabilization, the vesicles generally survive a single pressure cycle, as they are destroyed by contact with the supercritical carbon dioxide via the phase boundary. As soon as the aforementioned stabilization step is reached, the integrated peptide will significantly increase the lifetime of the vesicles. Under these circumstances, the stabilized vesicles will persist and accumulate. The stabilizing peptides will be analyzed and artificially reproduced. Using the synthetic peptides, the corresponding vesicles will be re-assembled and submitted to further structural analysis, including field gradient NMR spectroscopy. At this stage, the thermal stability of the optimized vesicles can be directly compared to the original vesicles without integrated peptides.

The principal capability of membrane vesicles for the early steps of life has been described on numerous occasions [27,28,29]. Recently, it was shown that a possible prebiotic amphiphile (decanoic acid) undergoes distinct binding interactions with nucleobases and proteins [30,31]. So, at this stage, they may serve as containments for initial metabolic steps and, most prominently, as an ensemble of encapsulated environments for the replication of RNA molecules.

## Figures and Tables

**Figure 1 life-07-00003-f001:**
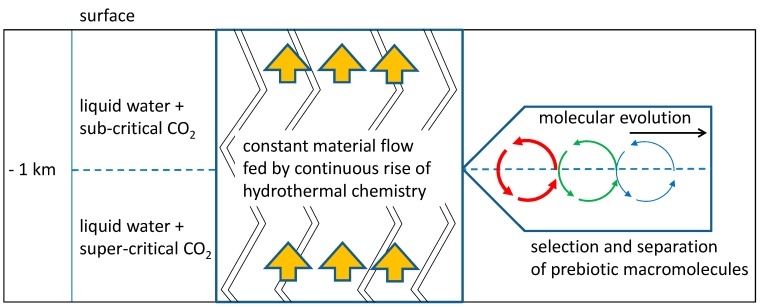
Schematic depiction of molecular evolution in a tectonic fault zone. A constant supply of hydrothermal products pass through a zone of vesicle formation (h = −1 km). In this “vesicle zone”, prebiotic molecules which integrate into vesicle membranes are selected and accumulated. A periodic process of vesicle formation presents the perfect scenario for subsequent molecular evolution (right).

**Figure 2 life-07-00003-f002:**
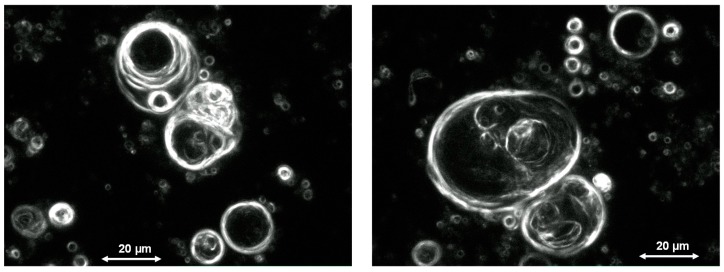
Vesicles formed in a high-pressure cell filled with a two-phase mixture of water and scCO_2_ containing small amounts of phospholipid as a model amphiphile. The pressure and temperature conditions as well as the composition of the bulk medium correspond to those expected in a tectonic fault zone in a depth of 1 km.

**Figure 3 life-07-00003-f003:**
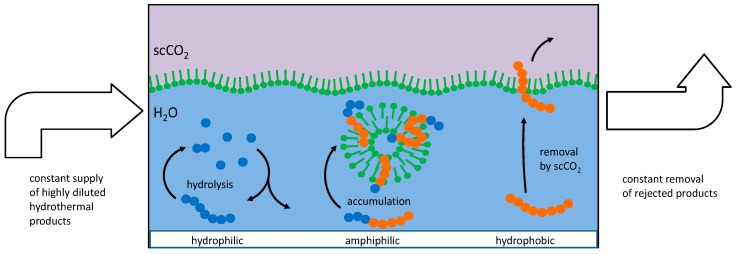
Mechanism of peptide selection and accumulation in the presence of vesicles. Left: Peptide chains formed by hydrophilic amino acids (blue circles) will undergo little interaction with vesicles and remain in the aqueous phase where they undergo hydrolysis. Right: Peptide chains formed by hydrophobic amino acids (red circles) will eventually be eluted by scCO_2_. Center: Amphiphilic peptides will accumulate in the bilayer membrane and remain partially protected against hydrolysis and elution.

**Figure 4 life-07-00003-f004:**
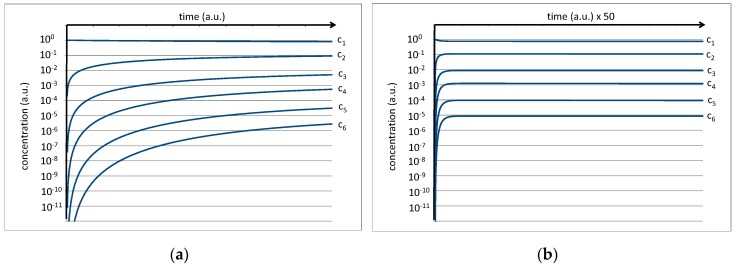
(**a**) Numeric simulation of the development of the peptide concentration assuming k_c_/k_h_ = 0.1 and a starting amino acid concentration of c_1_(0) = 1 a.u. The right image (**b**) shows the same result on a scaled time axis.

**Figure 5 life-07-00003-f005:**
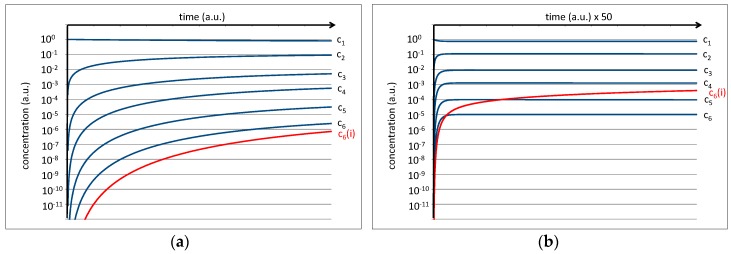
(**a**) Numeric simulation visualizing the integration of the amphiphilic fraction of hexapeptides (red line) into vesicle membranes. The right image (**b**) shows the same result on a scaled time axis.

**Figure 6 life-07-00003-f006:**
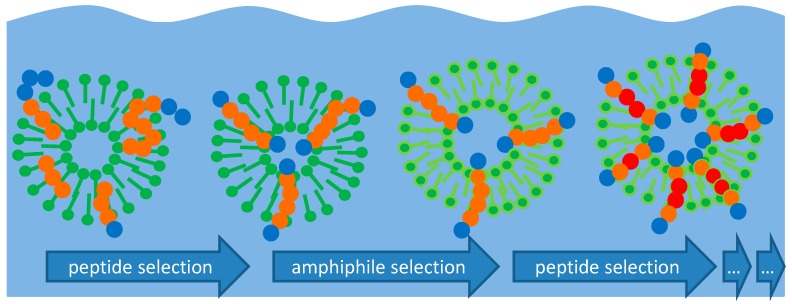
Symbolic depiction of a possible structural evolution of vesicles by repeated optimization steps. Peptide selection and integration (left) changes the criteria for the selection of other amphiphiles, leading to an alteration of the membrane composition (center) which in turn changes the criteria for peptide selection (right).

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
