# Peer review of "Selection of Prebiotic Molecules in Amphiphilic Environments"

_life, 2017, doi:10.3390/life7010003_

Round 1

Reviewer 1 Report

Mayer, Schreiber and Davila present a hypothesis on the prebiotic selection and accumulation of amphiphilic peptides. Their hypothesis consists of two parts. The first part deals with the relevant geological setting (tectonic fault zones) and the formation of vesicles there. In the second part a (partly quantitative) model for the formation and accumulation of peptides is outlined. The ideas presented in this manuscript are highly speculative but not implausible. Therefore, they are worth to be considered and experimentally tested. The numerical results shown in Figure 5 are impressive and indicate a possible prebiotic relevance of the model. However, I have some points of criticism.

Most of the first part of the manuscript is not new and has been published before in OLEB (Mayer, Schreiber and Davila, 2015). Moreover, many passages of the manuscript are even identical to passages in this earlier paper. Here are three examples:

Example #1:

The authors write (lines 86ff) „The sizes of the cavities range over several orders of magnitude (from sub-mm to several decimeters) and add to an overall volume of several cubic kilometers. Existing examples for such structures are low-temperature hydrothermal systems with strong gas flux (mofettes, CO2 geysers) in the continental crust. They contain supercritical and subcritical water with various degrees of salinity as well as supercritical and subcritical gases,...“

Compare Mayer, Schreiber and Davila (OLEB 2015): „The sizes of the cavities range over several orders of magnitude (from sub-mm to several meters) and sum to an overall volume of several cubic kilometers. Examples for such structures are low-temperature hydrothermal systems with strong gas flux (mofettes) in the continental crust. They contain supercritical and subcritical water with various degrees of salinity as well as supercritical and subcritical gases.“

Example #2:

The authors write (lines 92ff): „...offer periodically changing pressure and temperature conditions, variable pH-values, clay and transition metal-containing mineral surfaces, and a large number of other potent catalysts. Processes like the Fischer-Tropsch synthesis (facilitated by catalysts containing Fe, Co or Ni) or the formation of racemic amino acids are likely under the given circumstances...“

Compare Mayer, Schreiber and Davila (OLEB 2015): „...combines periodically changing pressure and temperature conditions, variable pH-values, clay and transition metal-containing mineral surfaces, and a large number of other potent catalysts. Processes like the Fischer-Tropsch synthesis (which works with catalysts containing Fe, Co or Ni) or the formation of amino acids (which can be stabilized by a mixture of pyrite, pyrrhotite and magnetite) are likely under the given circumstances...“

Example #3:

The authors write (lines 103ff): „...where supercritical CO2 occurs (at approximately -1 km), a precipitation zone is expected where organic constituents may accumulate at strongly increased concentration. Altogether, tectonic fault zones represent a very potent environment for pre-biotic chemical synthesis.“

Compare Mayer, Schreiber and Davila (OLEB 2015): „... where supercritical CO2 occurs, a precipitation zone is expected where organic constituents may accumulate at strongly increased concentration. Altogether, the tectonic fault zone may represent an ideal environment for pre-biotic chemical evolution.“

This extent of self-plagiarism is unacceptable.

I recommend publication of the manuscript after the following major revisions:

(i) Lines 83 to 145: Mainly a repetition from the authors‘ previous paper. Should be cut by half.

(ii) In some cases, statements are not sufficiently supported:
Figure 1: Add original references describing the occurrence of supercritical carbon dioxide in tectonic fault zones.
Lines 92 to 94: Original references needed on the periodically changing pressure and temperature conditions in tectonic fault zones.
Lines 262 to 266: The parameter values used in the calculation must be substantiated by data from the literature.

(iii) All self-plagiarism must be removed (not only in the three examples given above).

Author Response

We want to thank for the comments and recommendations of reviewer 1 and will respond to the four given proposals for revision one by one:

(i) Lines 83 to 145: This section is a repetition from a previous paper and should be cut by half.

In fact, this section was meant as a reproduction of a basic model in order to improve the readability of the article. However, this information can also be obtained by reading the original paper. In order to avoid extensive self-quotation, we have now shortened and re-worded this section as proposed. It is now free of any significant self-quotation.

(ii) Figure 1: Add original references describing the occurrence of supercritical carbon dioxide in tectonic fault zones.

We have added reference [15] at the corresponding point in the text.

Lines 92 to 94: Original references needed on the periodically changing pressure and temperature conditions in tectonic fault zones.

We have added reference [17] at this point.

Lines 262 to 266: The parameter values used in the calculation must be substantiated by data from the literature.

Unfortunately, there are no literature data available which are valid for the given temperature and pH conditions. The parameters settings rely on experimental data which we have collected so far. In the given experiment, we treated a mixture of amino acids with the simulated conditions of -1 km depth (70 bar, 350 K) until a thermal equilibrium had been reached. Under these circumstances, NMR data show an approximate relation between c1 and c2 of 10:1. In a kinetic interpretation of the equilibrium constant (and, in first approximation, ignoring other connected equilibria), this relation can be directly seen as the relation between kc and kh. We now have included this statement in the section where we define the simulation parameters:

“The equilibrium constant of a single peptide bond formation is then given by the relation between kc and kh. Preliminary experiments of amino acid mixtures treated in a high pressure cell (P = 70 bar, T = 350 K) and analyzed by NMR show that a value of kc/kh = 0.1 is plausible. This is expressed by the fact that the average concentrations of the amino acids c1 and the average concentration of dipeptides c2 in thermal equilibrium exhibit an approximate relation of 10 : 1 which, in a kinetic interpretation of the equilibrium constant, should also roughly correspond to the relation kc/kh.”

(iii) All self-plagiarism must be removed (not only in the three examples given above).

We have removed and changed all sections which we may have been used on other occasions.

Reviewer 2 Report

            The authors have identified a central problem in prebiotic chemistry – concentration of reactants – and highlighted mechanisms for achieving high concentrations.  A chief mechanism entails percolation of reactants through tectonic faults. 

            The authors should be more explicit about the temperature of the hydrothermal environment.  Miller famously said that cycling of the prebiotic milieu through the hydrothermal vents (>400 °C every 10 million years) would cause decomposition of the constituents. 

            The authors mention “preliminary experiments … that give a kc/kh ratio of 0.1”  I would ask the authors to describe the experiment in detail.  I don’t necessary quibble with the value obtained, but want to know how this was obtained, and which amino acids were employed, so I can think about whether this is meaningful and generally applicable. 

            The abstract mentions, in the last sentence, the RNA world.  I see nothing in the paper about the RNA world; instead, the focus is on peptides. 

            As a position paper, this article is satisfactory, and should be published in Life.  I might suggest that the authors attempt to list a set of testable hypotheses as part of the end of the paper.  What really are the big unknowns?

Author Response

We want to thank for the comments and recommendations of reviewer 2 and will respond to the four given proposals for revision one by one:

(i) The authors should be more explicit about the temperature of the hydrothermal environment.  Miller famously said that cycling of the prebiotic milieu through the hydrothermal vents (>400 °C every 10 million years) would cause decomposition of the constituents. 

This actually is an important point, as all prebiotic chemistry on the long run will undergo thermal decomposition when it is part of such a cycling process. However, this problem is not relevant for the given scenario. In the case described in our manuscript, all reactions occur under relatively mild conditions (approximately 310 to 350 K, as we have now stated at the beginning of chapter 4). All reaction steps rely on compounds which have been freshly formed de novo in the hydrothermal environment. The selection and evolution process then occurs based on these freshly formed monomer units under temperature and pressure conditions that allow for long-term stability of organic compounds. As soon as a component leaves this environment, it is lost for the described selection process anyway, so its further fate (possible thermal decomposition) does not affect the described molecular evolution by any means. It may even deliver new raw material, in this case one could even think of an active recycling process.

(ii) The authors mention “preliminary experiments … that give a kc/kh ratio of 0.1”  I would ask the authors to describe the experiment in detail.  I don’t necessary quibble with the value obtained, but want to know how this was obtained, and which amino acids were employed, so I can think about whether this is meaningful and generally applicable.

In the given experiment, we treated a mixture of amino acids with the simulated conditions of -1 km depth (70 bar, 350 K) until a thermal equilibrium had been reached. Under these circumstances, NMR data show an approximate relation between c1 and c2 of 10:1. In a kinetic interpretation of the equilibrium constant (and, in first approximation, ignoring other connected equilibria), this relation can be directly seen as the relation between kc and kh. We now have included this statement in the section where we define the simulation parameters:

“The equilibrium constant of a single peptide bond formation is then given by the relation between kc and kh. Preliminary experiments of amino acid mixtures treated in a high pressure cell (P = 70 bar, T = 350 K) and analyzed by NMR show that a value of kc/kh = 0.1 is plausible. This is expressed by the fact that the average concentrations of the amino acids c1 and the average concentration of dipeptides c2 in thermal equilibrium exhibit an approximate relation of 10 : 1 which, in a kinetic interpretation of the equilibrium constant, should also roughly correspond to the relation kc/kh.”

(iii) The abstract mentions, in the last sentence, the RNA world.  I see nothing in the paper about the RNA world; instead, the focus is on peptides.

In our manuscript, the RNA world is only mentioned at the very end. One intention of our publication is to open a perspective where our model (selection and evolution of vesicles equipped with peptides) can lead to a very potent basis for a subsequent development of the RNA world. The RNA world itself however is in fact beyond the focus of the manuscript. 

(iiii) As a position paper, this article is satisfactory, and should be published in Life.  I might suggest that the authors attempt to list a set of testable hypotheses as part of the end of the paper.  What really are the big unknowns?

This we believe we have done so in the last paragraph of the paper. Here we write:

“The result of this “evolution” process would be optimized, stable and maybe functional vesicles. Being thermally stable, they could survive over longer periods of time and could form versatile compartments for further developments in molecular evolution.”

In the near future, we plan on experimentally proving the evolution process as well as the occurrence of stabilized vesicles. At the present time, it is unknown if this process will actually occur. We now experimentally search for the three possible steps, as is described in chapter 6 of the paper:

a) integration, b) stabilization, c) function.

So if a list of testable hypotheses is being asked for, it is given by the full content of chapter 6.

Reviewer 3 Report

The manuscript entitled “Selection of prebiotic molecules in amphiphilic environments proposes an interesting scenario of molecular evolution which due to its fundamental significance requires thorough analysis. Specifically the authors in the last three lines in the abstract of their article support: “Altogether, the proposed scenario can be seen as an ideal environment for constant, undisturbed molecular evolution in and on cell-like compartments, the latter offering preferential starting conditions for a subsequent RNA world.”. This conclusion has therefore to be further analyzed in the main text of the article, discussing primarily the timing  of RNA World with that of the Lipid World.

Author Response

We want to thank for the comment and the recommendations of reviewer 3:

The manuscript entitled “Selection of prebiotic molecules in amphiphilic environments” proposes an interesting scenario of molecular evolution which due to its fundamental significance requires thorough analysis. Specifically the authors in the last three lines in the abstract of their article support: “Altogether, the proposed scenario can be seen as an ideal environment for constant, undisturbed molecular evolution in and on cell-like compartments, the latter offering preferential starting conditions for a subsequent RNA world.”. This conclusion has therefore to be further analyzed in the main text of the article, discussing primarily the timing of RNA World with that of the Lipid World.

We believe that we have some expertise in the details of the tectonic fault scenario as well as in lipid synthesis, vesicle formation and amino acid-peptide equilibria and corresponding kinetics. However, we do not see ourselves as experts in the RNA world theory. We therefore see our article as an offer to the RNA world community to use the idea of self-optimized vesicles (which already have been used in numerous RNA experiments) in combination with functional peptides as a suitable starting point. For this reason, we have included this key word in the abstract.

Regarding the timing issue, we think we have indicated in our last statements that we see a clear sequence: lipid world first, RNA world later. All other details should be open for discussion. In summary, the openness of our manuscript regarding the issue of the RNA world is intentional.

Reviewer 4 Report

This paper restates the authors' interesting hypothesis that tectonic fault zones containing supercritical CO2 played a critical role in prebiotic chemistry. The addition of clarifying new schematic figures, a summary of their 2015 paper on vesicle formation, a mathematical simulation of how peptides could accumulate, and a detailed discussion of vesicle-based evolution make this review a valuable extension of the previous work.

In some places the authors should clarify whether statements are based on data, and they should include a couple of additional references.

Specific issues:

1.  They suggest that these systems would be "constantly supplied with a huge variety of hydrothermally formed compounds from geological sources" (line 66, and elsewhere).  Have many organic compounds actually been found in such systems, and if so can their abiotic genesis be validated?  The authors should clarify whether there is any data to help answer these questions.

2.  Somewhat along the same line:  on line 95 they provide references for the likelihood that amino acids form under conditions found in tectonic fault zones, but no references for formation of fatty acids under such conditions.  

3.  Line 98:  "could support" rather than "supports".

4.  Line 101:  do the authors mean "in rising CO2" instead of "of rising CO2"?

5.  Line 108:  I suggest starting the sentence with "Recently we established a mechanism . . ."

6.  Line 122:  clarify why the vesicles would be "thermodynamically" unstable.  E.g., due to dilution of the lipids?

7.  Lines 157-161:  An example of such selection by vesicles has indeed been published and should be cited:  "Nucleobases bind to and stabilize aggregates of a prebiotic amphiphile, providing a viable mechanism for the emergence of protocells", Proc. Natl. Acad. Sci. USA 110, 13272-13276 (2013).

8.  Line  159:  should be "thereby"?

9.  Line 178:  do the authors mean "preliminary" experiments?

10.  Line 205:  "has" rather than "had"

11.  Line 288:  should be "thereby"?

12.  Line 291:  "negligible" would be better 

13.  Line 306:  "across" rather than "along"

14.  Lines 318-322:  This idea of evolving affinity of vesicles has also been proposed in another recent review, which should be cited:  "A self-assembled aggregate composed of a fatty acid membrane and the building blocks of biological polymers provides a first step in the emergence of protocells", Life 6, 33 (2016).

Author Response

We want to thank for the comments and recommendations of reviewer 4 and will respond to all given proposals for revision one by one:

1.  They suggest that these systems would be "constantly supplied with a huge variety of hydrothermally formed compounds from geological sources" (line 66, and elsewhere).  Have many organic compounds actually been found in such systems, and if so can their abiotic genesis be validated?  The authors should clarify whether there is any data to help answer these questions.

The formation of prebiotic molecules such as amino acids in hydrothermal vents has been proposed quite early (Tunnicliffe, V. (1991). "The Biology of Hydrothermal Vents: Ecology and Evolution". Oceanography and Marine Biology: An Annual Review. 29: 319–408. ). Hydrothermal sources also have been analyzed for prebiotic molecules, with mixed success. The problem is not the lack of prebiotic chemistry in submarine hydrothermal vents, on the contrary, one generally finds too many of them (see, for example “Amino acids in water samples from deep sea hydrothermal vents at Suiyo Seamount, Izu-Bonin Arc, Pacific Ocean” T. Horiuchi et al., Organic Geochemistry 35, 1121 (2004)). In the given case, most of these amino acids are of biogenic origin as shown by their stereochemistry: the majority is found as the L-enantiomer. However, racemic mixtures and non-biogenic amino acids have been detected as well which indicates their non-biogenic origin. The best evidence for the possible formation of amino acids under hydrothermal conditions was achieved by William Marshall when he experimentally reproduced the synthesis of eleven proteinogenic amino acids from inorganic precursors under elevated temperature and pressure conditions (W. Marshall, “Hydrothermal Synthesis of Amino Acids”, Geochimica et Cosmochimica Acta 58, 2099 (1994)). Similar results were achieved more recently in an energetic analysis by N. Kitadai (N. Kitadai, “Energetics of amino acid synthesis in alkaline hydrothermal environments”, Orig. Life Evol. Biosph. 45, 377 (2015)). So, to our opinion, there is more evidence for the hydrothermal formation of amino acids than against it. However, we feel that this discussion is beyond the topic of the given manuscript, therefore we would like to separate this from the issue which we want to actually deal with: The possible selection of prebiotic molecules under the given circumstances.

2.  Somewhat along the same line:  on line 95 they provide references for the likelihood that amino acids form under conditions found in tectonic fault zones, but no references for formation of fatty acids under such conditions.  
This in fact was a mistake on our part, we had positioned the wrong references at this point. However, since the corresponding statement now has been omitted (following the recommendation by reviewer 1), this problem has been removed.

3.  Line 98:  "could support" rather than "supports".

Thank you for this recommendation. Again: the statement has been removed by the shortening recommended by reviewer 1.

4.  Line 101:  do the authors mean "in rising CO2" instead of "of rising CO2"?

Thank you for this recommendation. Again: the statement has been removed by the shortening recommended by reviewer 1.

5.  Line 108:  I suggest starting the sentence with "Recently we established a mechanism . . ."
Thank you for this recommendation. We have changed the text accordingly.

6.  Line 122:  clarify why the vesicles would be "thermodynamically" unstable.  E.g., due to dilution of the lipids?

Vesicles are thermodynamically unstable structures per se (see, for example, K. Morigaki et al., “Thermodynamic and kinetic stability: properties of micelles and vesicles”, Colloids and Surfaces A 213, 37-44 (2003)). So they only survive due to kinetic stability. That is, the process of their decomposition may be slow enough to allow for long-term existence.

7.  Lines 157-161:  An example of such selection by vesicles has indeed been published and should be cited:  "Nucleobases bind to and stabilize aggregates of a prebiotic amphiphile, providing a viable mechanism for the emergence of protocells", Proc. Natl. Acad. Sci. USA 110, 13272-13276 (2013).
That is a very important point: thank you for making us aware of this publication. We have added this reference to our manuscript and introduced a corresponding statement in the discussion.

8.  Line  159:  should be "thereby"?

Thank you for this recommendation. We have changed the text accordingly.

9.  Line 178:  do the authors mean "preliminary" experiments?

Thank you for this recommendation. We have changed the text accordingly.

10.  Line 205:  "has" rather than "had"

Thank you for this recommendation. We have changed the text accordingly.

11.  Line 288:  should be "thereby"?

Thank you for this recommendation. We have changed the text accordingly.

12.  Line 291:  "negligible" would be better 

Thank you for this recommendation. We have changed the text accordingly.

13.  Line 306:  "across" rather than "along"

Thank you for this recommendation. We have changed the text accordingly.

14.  Lines 318-322:  This idea of evolving affinity of vesicles has also been proposed in another recent review, which should be cited:  "A self-assembled aggregate composed of a fatty acid membrane and the building blocks of biological polymers provides a first step in the emergence of protocells", Life 6, 33 (2016).

Again, we are grateful to the reviewer for having informed us on this publication. We have quoted it together with the abovementioned article at the end of our manuscript.

Round 2

Reviewer 1 Report

I have only one suggestion for a very minor correction:

Line 101: „the quality of the carbon dioxide as a solvent will drastically decrease“ should read „the quality of the carbon dioxide as a solvent will drastically change“ or „the ability of the carbon dioxide to act as a solvent will drastically decrease.“

In my opinion, the revised version is suitable for publication.

Author Response

We thank you for the recommendation. We have chosen the second version of the proposed text, the sentence now reads:

"At this point, the ability of the carbon dioxide to act as a solvent will drastically decrease and most of the less polar or amphiphilic organic constituents will precipitate."

Reviewer 3 Report

The answer of the authors to the reviewer is adequate and the manuscript can now be accepted for publication.

Author Response

We have followed your proposals to optimize our manuscript. Thank you again for the valuable recommendations.